# Prebiotics and the Human Gut Microbiota: From Breakdown Mechanisms to the Impact on Metabolic Health

**DOI:** 10.3390/nu14102096

**Published:** 2022-05-17

**Authors:** Cassandre Bedu-Ferrari, Paul Biscarrat, Philippe Langella, Claire Cherbuy

**Affiliations:** 1Micalis Institute, Institut National de Recherche pour l’Agriculture, l’Alimentation et l’Environnement (INRAE), AgroParisTech, Université Paris-Saclay, 78350 Jouy-en-Josas, France; cassandre.bedu-ferrari@inrae.fr (C.B.-F.); paul.biscarrat@inrae.fr (P.B.); philippe.langella@inrae.fr (P.L.); 2Yoplait France—General Mills, Vienne Technical Center, 38200 Vienne, France

**Keywords:** gut microbiota, prebiotics, carbohydrate metabolism, health and well-being, short-chain fatty acids, personalised nutrition

## Abstract

The colon harbours a dynamic and complex community of microorganisms, collectively known as the gut microbiota, which constitutes the densest microbial ecosystem in the human body. These commensal gut microbes play a key role in human health and diseases, revealing the strong potential of fine-tuning the gut microbiota to confer health benefits. In this context, dietary strategies targeting gut microbes to modulate the composition and metabolic function of microbial communities are of increasing interest. One such dietary strategy is the use of prebiotics, which are defined as substrates that are selectively utilised by host microorganisms to confer a health benefit. A better understanding of the metabolic pathways involved in the breakdown of prebiotics is essential to improve these nutritional strategies. In this review, we will present the concept of prebiotics, and focus on the main sources and nature of these components, which are mainly non-digestible polysaccharides. We will review the breakdown mechanisms of complex carbohydrates by the intestinal microbiota and present short-chain fatty acids (SCFAs) as key molecules mediating the dialogue between the intestinal microbiota and the host. Finally, we will review human studies exploring the potential of prebiotics in metabolic diseases, revealing the personalised responses to prebiotic ingestion. In conclusion, we hope that this review will be of interest to identify mechanistic factors for the optimization of prebiotic-based strategies.

## 1. Introduction

### 1.1. The Gut Microbiota as a Key Player in Human Health and Diseases

There is now a large body of evidence indicating that the gut microbiota plays a central role in human physiology and metabolism. Indeed, the gut microbiota supports the training and function of the host immune system [1,2], contributes to host metabolic homeostasis [3,4], influences neurocognitive function, and provides colonisation resistance to invasive pathogenic infections [5,6]. This gut microbial community is dominated by bacteria, which have been extensively studied in the past decades. The microbiota also includes commensal populations of fungi, viruses, and archaea that have not been as widely explored [7]. In healthy adults, over 90% of the gut bacterial diversity belongs to two phyla, Gram-positive *Firmicutes* and Gram-negative *Bacteroidetes* [8]. The *Bacteroidetes* phylum contains the major families *Bacteroidaceae*, *Prevotellaceae*, *Rikenellaceae*, and *Porphyromonadaceae*. Groups of *Firmicutes* are more diversified and, primarily affiliated with four classes, including, in decreasing order of relative abundance, *Clostridia*, *Bacilli*, *Erysipelotrichi*, and *Negativicutes*. Each is composed of several bacterial families, such as *Lachnospiraceae*, *Ruminococcaceae*, *Clostridiaceae*, *Christensenellaceae*, *Eubacteriaceae*, and *Peptostreptococcaceae* [9,10]. Less abundant *Actinobacteria*, *Proteobacteria*, and *Verrucomicrobia* represent subdominant bacterial phyla in humans. 

A growing number of chronic disorders, with distinct clinical presentations, share underlying patterns of major compositional changes in the gut microbiota, often referred to as “dysbiosis”. Thus, diabetes [11], cardiovascular disease [12,13], obesity [14], inflammatory bowel disease [15,16], and non-alcoholic fatty liver disease (NAFLD) are examples of diseases associated with a shift in microbial patterns compared to healthy individuals [17]. Such observations advocate for microbiota-targeted interventions to preserve human health and prevent or treat diseases. In this context, diet-based interventions are promising tools to modify the gut microbiota towards a favourable community structure [18]. 

### 1.2. Prebiotics: An Old Concept with Innovative Applications

The concept of prebiotics refers to the nutritional strategy that aims to fine-tune the composition and function of the gut microbiota to favour health and well-being. This concept was first introduced in 1995, and since then has evolved, along with advances in our understanding of the gut microbiota (Table 1 for the different definitions and their evolutions). The International Scientific Association for Probiotics and Prebiotics (ISAPP) proposed the most recent consensus view that redefines a prebiotic as “a substrate that is selectively utilised by host microorganisms conferring a health benefit” [19]. Prebiotics fulfil three criteria: (a) resistance to gastric acidity, hydrolysis by mammalian enzymes, and gastrointestinal absorption; (b) fermentation by intestinal microflora; and (c) selective stimulation of the growth and/or activity of intestinal bacteria associated with health and well-being [20]. A comprehensive understanding of the health properties of a substance classified as a prebiotic should require evaluating the physiological effect in randomized controlled human trials [21]. Thus, consistent clinical studies are needed to confirm the safety of prebiotics and their use in an appropriate dose. Historically, prebiotics were identified to stimulate *Bifidobacterium* and *Lactobacillus* bacterial groups, which are often referred to as probiotics [22]. This was based on the assumption that an increase in relative abundance of these bacteria could be used as a benchmark for a healthy gut microbiota, although the causal relationship between specific microbial changes and benefits for host health and well-being remains to be proven [23]. However, more recently, data based on high-throughput sequencing techniques, has revealed that the impact of prebiotic-induced changes on the gut microbiota is more widespread, probably due to functional redundancy and cross-feeding interactions [24]. Thus, for example, *Bifidobacterium* and *Lactobacillus* produce predominantly acetate and lactate, which can then stimulate a number of bacterial species benefiting from the presence of the prebiotic [25]. Therefore, the requirement that prebiotics must be “specific” or “selective” by a limited number of taxonomic groups/species or metabolic activities that promote health has been questioned, as it conflicts with the current understanding of gut microbiota ecology and its correlation with health [24]. The most recent ISAPP definition widened the concept of prebiotics beyond stimulation of bifidobacteria and lactobacilli and recognised that health benefits can derive from effects on other beneficial taxa [19]. Implicit in this definition is the idea that a prebiotic should not be broadly metabolised by the gut microbiota but should be selectively metabolised by health-promoting microorganisms. 

### 1.3. The Sources & Nature of Prebiotics

Derived from plants, non-digestible carbohydrate-based polymers are important sources of prebiotics, although non-carbohydrate substances have begun to emerge for their prebiotic potential, including polyunsaturated fatty acids and polyphenols [27,28,29]. Similar to carbohydrates, certain of these phytochemicals have a low bioavailability, which indicates that they may escape absorption in the small intestine. It was estimated that 5 to 10% of the total intake of plant polyphenols reach the colon, where they can be metabolised to various degrees by the gut microbiota [30]. As this has been recently reviewed, polyphenols will not be discussed further [31]. In the present article, we will focus only on complex carbohydrates that represent the best-studied class of prebiotic until now.

Non-digestible dietary carbohydrates, mainly dietary fibres and resistant starches, escape host digestion and reach the lower part of the gastro-intestinal tract, where they can supply the gut microbiota [32]. There is a body of evidence showing that diets, especially in Western countries, lack non-digestible substrates, which reduces the bacterial fermentation in the gut environment. The trend is toward a reduction in dietary fibre intake, with present consumption being the lowest recorded in human history [33]. The intake of dietary fibres was estimated at 15–20 g/day, which is below the daily recommendations of 25–35 g/day [34]. However, the amount of carbohydrates was shown to reach 50 g/day in countries where there is a high intake of whole-grain cereals, legumes, fruits, and vegetables [35,36]. Human nutrition trials show that dietary fibre fermentation represents about 75 to 90% in fruits and vegetables and 25 to 35% in whole grains [37]. Each dietary carbohydrate exhibits distinct structural characteristics, related to the length of the molecule, sugar moieties, the presence of substituents, the linkages, and the side-chain branching, which influence the microbial digestibility of dietary carbohydrates [38]. Variations in the accessibility of polysaccharides for fermentation are considered critical in shaping the gut microbial ecosystem. 

Complex oligo- and polysaccharides, also designated as glycans, constitute the most heterogeneous and abundant polymer biomolecules in nature. The primary classification of carbohydrates is based on chemistry, which is molecular weight, character of individual monomers, degree of polymerization (DP), and type of linkage (α or β), as agreed at the Food and Agriculture Organization/World Health Organization Expert Consultation in 1997 [39]. Among the first group of dietary ingredients, human milk polysaccharides (HMOs), naturally present in breast milk, were recognised for their ability to modulate intestinal health and well-being. These mixtures of oligosaccharides, in combination with glycoproteins and glycolipids, result mainly in a dominant abundance of *Bifidobacterium* species in breastfed infants, and, to a lesser extent, in an abundance of some *Bacteroides* and *Lactobacillus* species [40]. As a result, different oligosaccharides have been used to mimic the function of HMOs, such as fructo-oligosaccharides (FOS) and galacto-oligosaccharides (GOS). 

Besides the fructan group, short-chain FOS or oligofructose and long-chain inulin constitute a mixture of linear fructan chains composed of both fructosyl-glucose and fructosyl-fructose linked by β (2-1) glycosidic bonds [20]. Inulin-type fructans (ITF) are often manufactured for use in the food industry by extraction from plants, such as onions, Jerusalem artichoke, chicory root, and agave. FOS can be produced by partial hydrolysis of long-chain inulin using endo-inulinase enzymes or derived from sucrose by transfructosylation using β-fructosidase enzymes [41]. The large variety of inulin chemical structures and botanical origins result in a wide range of ITF mixtures with different functional effects. From a chemical perspective, DP distinguishes FOS mixtures, in which DP varies from 2 to 7 with an average DP of 4 sugar units, from longer inulin molecules. For instance, the inulin from chicory is composed of a mixture of oligomers and polymers in which DP ranges from 2 to 70 units with an average DP of 12. 

There are two common types of GOS, both composed of galactose residues. Derived from lactose syrup, β-linked GOS, also known as trans-GOS (tGOS), are commercially synthesised by the transglycosylation activity of bacterial β-galactosidase or β-glycosidase enzymes [42]. This process commonly produces mixtures of short-chain tri- to penta-saccharides, which present a terminal glucose monomer and a wide variety of β (1-6), β (1-2), β (1-3), and mostly β (1-4) linkages. Likewise, α-linked GOS is another type of non-digestible fibre commonly found in plants, consisting of α-linked galactose, α-linked glucose, and a terminal β-linked fructose. They are commonly called raffinose family oligosaccharides (RFOs), including raffinose, stachyose, and verbascose. 

ITF and GOS represent the main confirmed prebiotics [18]. However, many other classes of non-digestible carbohydrates derived from plants can be considered as potential prebiotics because they reach the large intestine and are broken down by the gut microbiota [20]. This is the case for starch, one of the most popular nutritional sources for humans, for which a fraction can be resistant to host enzymes. Indeed, besides the digestible starch, which is rapidly or slowly hydrolysed, a variable fraction called resistant starch (RS) resists digestion in the small intestine and is fermented in the large intestine, where it provides nutrients for the gut microbiota. The non-digestible starch fraction contains a mixture of two major components, amylose and amylopectin, and other polysaccharides such as α-glucans [43]. RS has been well documented for its promising nutritional interventions in cardiovascular disease and in a variety of metabolic disorders [44]. Many other sets of complex polysaccharides are also promising potential prebiotics. Among them, xylo-oligosaccharides (XOS)/arabino-xylo-oligosaccharides (AXOS) are abundant in cereals and are mainly composed of a chain of xylose units linked by β (1-4) bonds with branching sugar residues. Several clinical studies show that XOS/AXOS could exert protective effects on intestinal homeostasis and metabolic status [45]. Pectins are important cell-wall components of many fruits and vegetables that are composed of a chain of α (1-4)-linked D-galacturonic acid units with side chains containing arabinans, galactans, and arabinogalactans [18]. β-glucans are found in wheat, oat, and barley and are composed of D-glucose monomers joined by mixed-linkages β (1-4) and β (1-3) of glucose residues. Consumption of pectins and β-glucans has shown beneficial effects on metabolic homeostasis in humans [46]. In view of the richness of polysaccharides and the diversity of yet unexploited plant sources, the variety of compounds with prebiotic potential should grow in the coming years [18].

## 2. The Gut Microbiota: An Ecosystem Designed for Carbohydrate Breakdown

### 2.1. CAZymes, Abundant Enzymes in the Gut Microbiota

Several studies have highlighted the gut microbiota to be one of the most specialised and sophisticated ecosystems involved in the breakdown of complex polysaccharides [47]. Indeed, intestinal bacteria display efficient strategies for breaking down carbohydrates in this dynamic nutrient environment [48]. Providing mechanisms of prebiotic utilisation is a step toward an efficient intentional modulation of the gut microbiota to improve human health and wellbeing [24]. In this context, prebiotic research is focusing on understanding how commensal microorganisms selectively scavenge carbohydrate substrates and act on their degradation and consumption [21]. In particular, certain gut microorganisms have repeatedly been shown to improve human health, and it is important to know how diet could be used to fine-tune the composition of these gut microbes, selectively stimulating the abundance of the health-promoting faction.

The functionality of the gut microbiota to break down complex carbohydrates is reflected in an arsenal of prominent and highly diverse carbohydrate active enzymes (CAZymes)-encoding genes, which comprise 1 to 5% of the predicted coding sequences in most bacterial genomes [49]. Thus, the gut microbiome displays a plethora of CAZymes dedicated to the breakdown, biosynthesis, and modification of complex carbohydrates. In contrast, the CAZyme repertoire encoded by the human genome is minimal (approximately 17 enzymes) and is restricted to the ability to degrade digestible starch, sucrose, and lactose, revealing that gut microorganisms have evolved into large enzymatic repertoires that expand human digestive physiology [50]. 

It is noteworthy that certain colonic bacteria are able to metabolise a remarkable variety of substrates, whereas other species have more restricted metabolic capacities. In particular, as members of the Gram-negative *Bacteroidetes* phylum, the *Bacteroides* genus exhibits a wide diversity and a large abundance of CAZymes-encoding genes. This group of bacteria appears to be one of the major actors in the breakdown of polysaccharides, which are metabolised into a form that could otherwise not be absorbed and/or utilised by the host. Due to this vast genetic repertoire encoding CAZymes, *Bacteroidetes* utilise a large number of different carbohydrate structures and so are often known as “generalists” [51]. Devoted to carbohydrate metabolism, the abundance of CAZymes in *Actinobacteria* members represented up to 8% of the *Bifidobacterium* genomes [52]. The CAZyme genetic content of *Firmicutes* has received less attention, although this phylum accounts for about 70% of the microbial diversity within the colonic microbiota of adult individuals [8]. Typically, at the level of bacterial phyla, *Firmicutes* genomes show fewer genes (mean 39.6) involved in the polysaccharide hydrolysis than *Bacteroidetes* spp. (mean 137.1) [50]. This is likely to be due, partially, to their smaller genome sizes and reflects their greater nutritional specialisation. Therefore, *Firmicutes* members are known as “specialists”. Some Gram-positive *Firmicutes* bacteria have been postulated to serve as “keystone” polysaccharide degraders, as their absence can limit the breakdown of a particular carbohydrate [53]. Interestingly, *Roseburia intestinalis* is unique among other *Firmicutes* studied, as strains of this species encode about 100 to 150 CAZymes [54].

CAZymes constitute an enzyme superfamily that has been classified into groups based on their amino acid sequence similarities and their biochemical reactions [49,55]. Diverse classes represent the numerous CAZyme families: glycoside hydrolases (GH) depolymerize carbohydrate substrates by hydrolysing glycosidic linkages; glycosyltransferases (GT) lead the formation of glycosidic bonds; carbohydrate esterases (CE) catalyze the hydrolysis of carbohydrate esters; and polysaccharide lyases (PL) operate the non-hydrolytic cleavage of glycosidic bonds. In addition, enzymes with auxiliary activities, including redox enzymes and carbohydrate-binding modules (CBM), are also involved in carbohydrate metabolism [56]. Functional prediction based on bacterial genomes revealed that GHs are the most representative CAZymes in the human gut microbiome [50]. The exploration of GH diversity highlighted the number of families increasing steadily, and as of March 2022, 173 sequence-based families of GHs have been defined in the continuously updated CAZy database [55]. 

Based on amino acid sequences, the CAZy classification has proven to be particularly robust for the prediction and characterisation of CAZyme activities [49]. Key active-site residues, catalytic mechanism, and the overall three-dimensional fold are strictly preserved, allowing the investigation of substrate specificity, which is a key facet of microbial responses to dietary carbohydrates [57]. Nevertheless, the broad diversity of sequences illustrates the difficulty in differentiating between family membership with distinct activity profiles and substrate specificity [38]. Indeed, CAZyme families are often “polyspecific” and include enzymatic activities with variations in glycosidic linkage specificity. Individual gut enzymes may hypothetically be associated with the breakdown of multiple polysaccharides that leads to ambiguous functional predictions [58]. 

### 2.2. Polysaccharide Utilisation Loci (PUL): The Paradigm of Bacterial Foraging Systems 

The complexity and diversity of the dietary carbohydrate structures that reach the lower digestive tract implies multi-step bacterial breakdown to efficiently utilise these macromolecules. Thus, nutrient acquisition strategies at the molecular level must be coordinated to enable commensal bacteria to thrive in this rich yet competitive intestinal environment. As primary degraders of polysaccharides in the gut, these orchestrated carbohydrate utilisation pathways were first investigated in *Bacteroidetes* species [59]. It has been shown that, to varying extents among *Bacteroidetes* species, bacterial genomes conserve a breakdown and importing machinery that is encoded within clusters of contiguous and coregulated genes, known as polysaccharides utilisation loci (PUL) [60]. This archetypal PUL, known as the starch utilisation system (SUS), encodes all components necessary for starch metabolism, including carbohydrate binding, breakdown, transport, and sensing proteins (Figure 1) [61,62]. Necessary to convert extracellular polysaccharides into intracellular monosaccharides, the SUS system-encoding genes are usually associated with three surface glycan-binding proteins (SGBPs), SusD, SusE, and SusF, to recruit the starch to the cell surface, where it is cleaved into malto-oligosaccharides by the outer-membrane-localized CAZyme, SusG. Subsequently, these latter are transported into the periplasm via the TonB-dependent transporter (TBDT), SusC, where they are broken down further into glucose, by the linkage-specific CAZymes, SusA, and SusB. The key inner-membrane-bound regulatory protein, SusR, controls the expression of the susA-G genes in response to the presence of malto-oligosaccharides in the periplasm [47,60]. Sequential susC/susD homologous genes constitute the signature of the canonical PUL machinery. These hallmarks have been used to explore PUL components among the genomes of *Bacteroidetes* members, and this exploration revealed the effective nutrient adaptation of *Bacteroidetes* bacteria [60,63]. For instance, the genomes of *B. thetaiotaomicron* and *B. ovatus* contain about the same number of PUL (about 100), of which few are common to both species, suggesting that these two symbionts can have distinct glycan niches [60].

The development of automatic bioinformatics by the CAZy team has fuelled the PUL database, which catalogues all experimentally characterised PUL from *Bacteroidetes* species [57,64]. Along with genomic, transcriptomic, structural biology, and protein biochemistry studies, a tremendous diversity of PUL, targeting specific plant, algal, animal, and microbial glycans, has been identified and functionally characterised. Thus, the gut microbiome exhibits extensive repertoires of PUL that underlie divergent carbohydrate specificities and functional adaptation to break down a wide variety of carbohydrate substrates [65,66].

The PUL systems constitute the major strategy for harvesting carbohydrates deployed by *Bacteroidetes* bacteria. Nevertheless, analogs of *Bacteroidetes* PUL have been described beyond the archetypal tandem susC/susD-like pairs [67]. Although distinct in structure from Gram-negative systems, prominent *Firmicutes* and *Actinobacteria* members elaborate additional cell surface-associated systems for the utilisation of polysaccharides, which are often built around enzymes with catalytic domains, similar to those of *Bacteroidetes* [48]. Substrate-specific gene clusters targeting a variety of plant- and host-based glycans were identified in the genomes of *Eubacterium rectale* and *Roseburia* species as Gram-positive PUL (gpPUL) [54,68,69]. The gpPUL contain carbohydrate transport systems are mostly adenosine triphosphate (ATP)-binding cassette (ABC), major facilitator superfamily (MFS), and phosphoenolpyruvate-phosphotransferase system (PTS) transporters. These genes are colocalised with associated lactose repressor (LacI) or cytosine arabinoside (AraC)-like transcriptional regulators and a minimum of one CAZyme [70,71]. Some intestinal Gram-positive bacteria have adapted another extracellular multienzyme complex, known as a cellulosome, which targets cellulose, resistant starch, and possibly other substrates. This glycan acquisition paradigm of Gram-positive species has been well characterised in carbohydrate-degrading microorganisms from the bovine rumen and soil [48]. Genomic, biochemical, and molecular approaches of the human gut microbiome revealed the presence of protein components (dockerins and cohesins) that are signatures of cellulosomes in colonic *Ruminococcus* bacterial species [72]. Another strategy constitutes the lytic polysaccharide monooxygenases (LPMOs), which rely on lignocellulose degradation. Classified as enzymes for auxiliary activities in the CAZy classification [49], LPMOs have not been identified to date in any human microbiome data, presumably due to the anaerobic environment present in the distal human intestine [73]. 

### 2.3. Nutrient Acquisition Strategies by the Microbial Community

Members of the gut microbiota do not live in isolation but are part of a dynamic community where different types of interactions and nutrient resources exist. As specified below, *Bacteroidetes* are functionally diversified to assimilate a wide range of complex polysaccharides that reach the distal part of the intestine. Two common strategies for nutrient acquisition have been described in the complex gut microbial ecosystem: one is selfish and the other is cooperative sharing [66] (Figure 1). The degree to which carbohydrate degradation is mediated by syntrophic interactions between different members of the gut microbiota, as well as the ratio between selfish and cooperative strategies, are unclear. In some altruistic *Bacteroidetes* species, polymers and hydrolytic enzymes seem to act as purveyors of beneficial public resources in various networks of polysaccharide utilisers, which benefit the whole community [74]. Indeed, harbouring signal sequences, hydrolases, and proteases are preferentially packed in outer membrane vesicles (OMVs) and are further released into the extracellular environment. Most of these PUL-derived enzymes are OMV surface-exposed, aiding bacteria incapable of efficiently using complex polysaccharides [75]. Hence, ubiquitous among Gram-negative bacteria, including *B. ovatus*, *B. fragilis*, and *B. thetaiotaomicron*, vesiculation appears to optimise the breakdown of nutrients by other bacteria with different metabolic capacities [76]. Most of these OMV-enriched-enzymatic activities contributes to the stability of the gut microbiota [77]. Leaving the rise of metabolic cross-feeding, some primary *Bacteroidetes* degraders diffuse simple oligosaccharides produced at the cell surface into the extracellular environment. Releasing polysaccharide breakdown products, this PUL-based distributive mechanism concomitantly benefits the bacterial community, which lacks the enzyme machinery to process the initial depolymerization step [78]. For instance, the large repertoire of GH/PL of *B. ovatus* appears more likely to break down a wide range of polysaccharides giving rise to the production of nutrients, which can help thrive other members of the community [77]. Similarly, it has been shown that *B. cellulosilyticus* initiates carbohydrate degradation through a surface endo-glycoside hydrolases, and releases oligosaccharides that are further metabolised by other members of the gut microbiota [79,80]. Thus, members of the *Bacteroidetes* act as primary polysaccharide degraders in orchestrating the initial polysaccharide breakdown that makes nutrients available not only for their own metabolic processes, but also to cross-feed the members of the gut microbiota depleted of hydrolytic enzymes.

This cooperative evolution of mutualistic interactions in *Bacteroidetes* members results in positive effects on bacterial fitness. It is noteworthy that sharing common resources depends on the structural complexity of carbohydrates that influences their microbial accessibility [78]. Based on carbohydrate structure, the symbiotic interaction network reflects the degradative hierarchies of carbohydrate utilisation, in which some simple glycans are prioritized above more complex polysaccharides [81]. In fact, individual bacteria exhibit multiple prioritization strategies in the presence of competing carbon sources. The simultaneous presence of dietary glycans, particularly in the form of monosaccharides, participates in consistent repression of the breakdown of host-derived glycans in *B. thetaiotaomicron* [82]. In general, the presence of highly prioritized glycans represses the transcription of genes involved in utilizing nutrients of lower priority [83]. Hierarchical orders promote the coexistence of stable microbial communities in a competitive environment [84]. The selective metabolism of substrates supports the dynamics of microbial communities and promotes the metabolic plasticity of the entire gut microbiota, which is constantly faced with nutritional changes. 

Interestingly, carbohydrate metabolisation does not always occur in the extracellular environment. Indeed, CAZymes on the cell surface can actually generate large oligosaccharides. Rapid transportation of PUL-produced oligosaccharides into the periplasm of Gram-negative bacteria may minimize substrate availability to other microbial residents. This “selfish” strategy prevents other bacterial species using partially degraded products [85]. Exhibiting little collaboration during the digestion of complex carbohydrates, *B. thetaiotaomicron* rapidly imports some glycans into the periplasm for further breakdown, conferring, at this step, no direct benefits on neighbouring species and enabling the microorganism to thrive in the competitive environment. Nevertheless, the metabolic activities of this bacterium can generate end-products that fuel other members of the gut community. These “partial” selfish ecological and evolutionary behaviours drive interdependent patterns for bacterial fitness and reciprocal feedback benefits of the entire gut microbiota. This selfishness strategy is central to the eco-evolutionary stability of extracellular polysaccharide digestions between microbial species. Bacterial interactions tend to significantly benefit the fitness of the overall microbial environment through the promotion of cooperation rather than competition [74,86]. The coordination of carbohydrate utilisation systems represents an impressive evolutionary solution for capturing valuable carbon sources. The rich interaction network between carbohydrates and bacteria is highly complex on the scale of an entire gut microbiota. 

Strategies for carbohydrate metabolisation should be understood both at the level of individual microorganisms and at the level of the intermingled microbial community [67,87]. Carbohydrate metabolisation varies according to the nutrient acquisition strategies, leading to different interdependent catalytic mechanisms that preferentially support the growth of certain microbes. The preferential degradation of some glycans over others is likely to play a central role in the complex relationships of the gut microbiota [88]. Understanding the glycan utilisation of intestinal bacterial is essential in bringing about changes in the intestinal microbiota with the aim of improving health through diet. Predicting individual strategies of nutrient acquisition for each microbial type completes the characterisation of microbial adaptation to prebiotics to improve human metabolism [89].

### 2.4. Short-Chain Fatty Acids (SCFAs): Key Metabolites Underpinning the Prebiotic Impacts 

Since the gut microbiota degrades complex dietary carbohydrates that cannot be broken down by human enzymes into digestible nutrients, it can be considered as “an energy harvesting” system. Conditions in the colon are highly favourable for complex carbohydrate fermentation due to the anaerobic environment, low transit time, and low pH coupled with low redox potential. Fermentation activities vary along the gastrointestinal tract according to dietary residues. In the proximal part of the colon, bacterial growth and fermentation are high, owing to the greater substrate availability. In the distal colon, the levels of substrates progressively decrease [90]. Several factors influence fermentation levels, including the shape and size of the food particles, the ratios of macronutrients, and transit time. Depending on the provision of adequate substrates, gut bacteria generate metabolites that can be quickly absorbed by the intestinal epithelium [91]. 

By-products derived from dietary carbohydrate fermentation are primarily SCFAs, mainly acetate, propionate, and butyrate [92]. Depending on the fibre content in the diet, the composition of the microbiota and the transit time of the gut, SCFAs typically reach total concentrations of 50 to 200 mmol/kg of luminal content in the human large intestine [93]. Acetate, propionate, and butyrate are typically found in a proportion of 3:1:1 in the gastrointestinal tract [94]. As detailed below, there are distinct pathways of bacterial fermentation that result in the production of short-chain fatty acids (SCFAs) involving different intermediate metabolites such as pyruvate, succinate, lactate, 1,2-propanediol, and acetyl-coenzyme A (coA) (Figure 2) [25]. In addition, small but significant amounts of alcohols, including ethanol, propanol and 2,3-butanediol can be formed as end-products of carbohydrate fermentation [25]. Furthermore, gaseous by- and end-products, such as hydrogen (H_2_), carbon dioxide (CO_2_), and sulphate (SO_4_^2−^) propel fermentation forward. Although energy needs are met primarily through fermentation, anaerobic respiration, through the membrane electron transport chain, confers the capacity to use H_2_ and CO_2_ by reductive acetogen and methanogen microorganisms, respectively, and SO_4_^2−^ by sulphate-reducing microorganisms, as electron acceptors [95]. The utilisation of these gaseous substrates is mainly the result of cross-feeding between members of the gut microbiota, rather than host absorption [96]. 

Among the bacterial metabolites produced by anaerobic fermentation of carbohydrates, SCFAs have been shown to exert multiple beneficial effects on host physiology. They represent a major source of energy for the host, in general, and the colon, in particular [97,98]. In humans, these bacterial metabolites provide about 10% of the daily caloric requirements. Importantly, butyrate is the most preferred energy supply for colonocytes and is used in the Krebs cycle and in ketone bodies production pathway [99]. Observations in germfree condition have highlighted the important role of microbiota-derived butyrate in colonocyte feeding. Indeed, in this condition, where there is no butyrate production, a colonic mucosal atrophy is observed and is associated to metabolic starvation and a deficit in mitochondrial respiration [100,101,102]. Furthermore, the intestinal microbiota has been shown to contribute to the metabolic specialisation of the colonic epithelium with the use of butyrate, and in vivo data has shown that butyrate can modulate the expression of an enzyme involved in its own metabolism [103]. Butyrate also maintains the intestinal barrier integrity by promoting cell proliferation, apoptosis, tight junctions, and mucus production [104]. Once produced in the intestinal lumen, acetate and propionate are easily absorbed by the colonic wall. They enter the portal blood compartment and are preferentially metabolised in the liver. They can fuel hepatic biosynthesis pathways, such as gluconeogenesis, cholesterol, and long-chain fatty acids synthesis, thus contributing to the whole host metabolism [105]. SCFAs can also act as microbial-signalling molecules that are recognised by specific host receptors, such as G-protein-coupled receptors, including GPR41 (also known as Free fatty acid receptor 2 (FFA2)), GPR43 (also known as Free fatty acid receptor 3 (FFA3)), GPR109A (also known as Hydroxycarboxylic acid receptor 2 (HCA2)), and Olfactory receptor 78 (Olfr78) [106,107]. In particular, GPR41 and GPR43 are expressed in entero-endocrine cells where they can activate a series of events conferring an array of metabolic effects, including the production of glucagon-like peptide-1 (GLP-1) and the appetite-regulating hormone peptide YY (PYY) [108]. Furthermore, the wide range of expression of these receptors, including in immune cells, suggests several other potential functions of SCFAs, especially those involving immune response and inflammation [109]. 

Given the role of these bacterial metabolites in human health, a detailed understanding of the metabolism of SCFAs by the gut microbiota is required to optimise efficient diet modulation strategies. SCFAs differ in relative production rates, concentrations, and bacterial producers. Metagenomic approaches have facilitated the characterisation of bacteria responsible for SCFA production (Figure 2) [110,111]. Found in the highest concentrations in the gut lumen, acetate is mainly produced by the decarboxylation of pyruvate. This step is followed by the hydrolysis of acetyl-CoA to acetate by an acetyl-CoA hydrolase. Playing an essential role in the central metabolism, most acetate is synthesised through this pathway by gut commensal bacteria, including *Prevotella* spp., *Ruminococcus* spp., *Bifidobacterium* spp., *Bacteroides* spp., *Clostridium* spp., *Streptococcus* spp., *A. muciniphila*, and *B. hydrogenotrophica*. Different subsets of gut bacteria are involved in the production of propionate and butyrate [25]. Propionate biosynthesis involves three different biochemical pathways, namely succinate, acrylate, and propanediol routes that are distributed within different bacterial groups [111]. Propionate synthesis through the succinate pathway appears to be the main route in the gut and is found primarily in bacteria belonging to the *Bacteroidetes* phylum and in a few members of *Firmicutes* [25]. Interestingly, faecal propionate levels (as a percentage of all fermentation acids) have been reported to be significantly correlated to their relative abundance of *Bacteroidetes* [112]. The acrylate and propanediol pathways concern a reduced number of gut bacteria. Some members of the *Lachnospiraceae* family, including *Roseburia inulinivorans* [113], as well as *A.*
*hallii*, *Blautia* species, and *A. muciniphila* can synthesise propionate through the propanediol pathway [114,115]. This involves the formation of 1,2-propanediol from deoxy sugars, such as rhamnose and fucose, and then its conversion into propionaldehyde and propionyl-CoA, leading to propionate. Consisting in the conversion of lactate to propionate, the acrylate pathway is the minor route of propionate synthesis and has been described in a small number of species such as *C. catus* [111]. Identification of butyrate-producing pathways revealed that they are mostly present in several different classes within the *Firmicutes* phylum. Indeed, butyrate producers are abundant in the gut and are primarily members of *Lachnospiraceae* and *Ruminococcaceae* [111]. For example, *Faecalibacterium prausnitzii* (which can represent more than 5% of the total bacterial population) [116], *Eubacterium rectale*, and *Roseburia* spp. (estimated to be 2 to 15% of the total bacteria within the *Lachnospiraceae*) are dominant butyrate-producing species in the human gut microbiota [117]. Butyrate production first involves the condensation of two molecules of acetyl-CoA to form butyryl-CoA through different steps, and then the liberation of butyrate from butyryl-CoA through two different types of reaction. In the first one, two enzymes are responsible for the conversion of butyryl-CoA into butyrate, i.e., phosphotransbutyrylase and butyrate kinase enzymes. In the second pathway, butyryl-CoA: acetate CoA-transferase converts butyryl-CoA into butyrate. Few bacteria in the gut use the butyrate kinase pathway, which seems to be limited to some *Coprococcus* species, whereas the butyryl-CoA:acetate CoA-transferase pathway is used by the majority of known butyrate-producing gut strains, such as *F. prausnitzii*, *E. rectale*, *E. hallii*, and *R. bromii* [118]. 

SCFA levels are frequently correlated to health. Indeed, metagenomic analyses, carried out on the butyrate production pathways, have shown that in diseases such as obesity and diabetes, the abundance of butyrate-producing species is lower than in healthy microbiota [119,120]. Interestingly, the amount of ingested dietary fibre has been shown to modulate the relative abundances of taxa and microbial genes associated with butyrate production. A low-fibre diet has been linked to reduced relative abundances of butyrate-producing microbes in humans [121], non-human primates [122], and mice, an effect which is more pronounced from generation to generation [123].

## 3. From Fundamental Research to Therapeutic Use

### 3.1. Prebiotics in Human Nutrition

Translating the fundamental research on the gut microbiota into clinical practice and/or interventional strategies constitutes a great challenge to improve public health. It is now time to revisit the prebiotic concept and deliver physiological benefits with rationally designed interventions based on gut microbiota engineering [124]. 

Recent human trials have shown the beneficial effects of prebiotics on the gut microbiota, health, and well-being. Given the nutritional impact of prebiotics on the gut microbiota composition and metabolic activities, this clinical data should be taken into account in future guidelines and public health recommendations [125]. Nevertheless, when considering the prebiotic impact on the composition and functions of the gut microbiota, several factors contribute to the gap of knowledge for the translation of prebiotics [19]. For example, functional effects of prebiotics depend upon the biological source that defines their chemical structures (e.g., molecular weight, individual monomers, degree of polymerisation, and type of linkage). This influences their fermentation by the gut microbiota and can contribute to a variety of physiological effects on health and wellbeing. Also, the nutrient composition of natural plant sources creates complex food matrices for microbes, which can impact the accessibility of polysaccharides to the gut microbiota. Finally, different types of polysaccharides co-exist in foods. For example, barley and oat cereals contain beta-glucans, arabinoxylans, starches, insoluble fibres, proteins, and other bioactive phytochemicals [126,127,128]. Thus, the advantage of the natural mix of different substances with bioactive properties should be better considered in intervention studies to ensure the utilisation by the gut microbiota functions prone to act on host physiology [125]. 

In addition, significant advances in the understanding of interactions between the gut microbiota, nutrition, and the host have not yet been considered in current public health recommendations. In particular, dietary fibre intake recommendations around the world refer to a certain amount of total dietary fibre for optimal health, typically 25 to 35 g/day [34]. However, these dietary guidelines do not consider the ratio of soluble versus insoluble dietary fibre, the levels of phenolic compounds, the fermentability by the colonic microbiota (rapid, slow, completely fermentable or to a limited degree), and the problems of digestive discomfort related to the consumption of dietary fibres. The balance between digestive tolerance and metabolic issues should be addressed in future dietary recommendations. There is a need for rigorous randomised intervention studies in humans with strict dietary control to evaluate the consumption of the types and amounts of prebiotics and their efficacy in the management of metabolic-related health issues [46].

### 3.2. Clinical Evidence of Prebiotic Effects

Several studies have already investigated the beneficial potency of ITF, including the two best characterized prebiotics inulin and FOS. In healthy adults, it has been shown that the consumption of chicory inulin regulates post-prandial glycemia and insulinemia. In a long-term dietary fibre supplementation, an acute dose of 24 g inulin in men with obesity decreased postprandial plasma glucose and insulin [129]. In overweight subjects, 20 g per day of inulin for 42 days decreased plasma insulin and improved insulin sensitivity [130]. Furthermore, ITF supplementation with a daily dose of 10 g for a duration of 6 weeks improved glycemic control for prediabetes and type 2 diabetes [131]. Similar effects were reported in healthy adults who consumed food products enriched in oligofructose [132]. 

In addition, inulin contributed to satietogenic effects through the suppression of ghrelin and increase of circulating GLP-1 and PYY concentrations, leading to the overall reduction in energy intake [133]. The improvement of appetite control was also reported in overweight and obese adults and children receiving a daily dose of 8 g for 16 weeks [134]. In overweight or obese adults, 21 g per day of oligofructose supplementation for 12 weeks attenuated body fat mass and weight gain. This was associated with a decrease in food intake and observed without changes in physical activity or lifestyle [133,134,135]. 

It has also been reported that a 30 day-inulin supplementation improved gastrointestinal disorders and inflammatory markers in middle-aged subjects [136]. The effect of inulin consumption on stool frequency was confirmed in healthy adults with mild constipation [137]. One potential mechanism for the beneficial effects of FOS could be through the improvement of obesity-associated intestinal permeability and inflammation. Indeed, a reduction of faecal markers related to intestinal integrity and inflammation was observed in obese patients after a prebiotic intervention consisting of a daily dose of 16 g of inulin for 3 months, in addition to dietary caloric restriction [138]. 

Inulin supplementation improved lipid metabolism in the management of dyslipidaemia associated with obesity and cardiometabolic risk. Physiological metrics identified signature biomarkers related to vascular, inflammatory, and lipid dysmetabolic morbidities (obesity and type 2 diabetes) in middle-aged individuals [139]. 

Furthermore, oligofructose supplementation in human studies induced specific microbial modifications that were associated with an increase in *Bifidobacterium* and *Bacteroides* abundances [130,138]. Indeed, specific inulin-induced changes in relative abundances of *Anaerostipes*, *Bilophila*, and *Bifidobacterium* were identified in 4-week intervention periods of 12 g of inulin consumed on a daily basis [137]. These microbial changes might result in a rise in SCFA production, revealing an increase in carbohydrate fermentation [125]. Thus, oligofructose was reported to increase the concentrations of acetate, propionate, and butyrate in faecal samples of healthy adults [137,140]. However, in contrast, other studies did not report any effect of inulin supplementation on SCFA levels [141]. Given these contradictory results, more evidence is needed to further elucidate the effect of inulin on SCFA production. Indeed, these bacterial-derived metabolites are key players involved in the metabolic health and diseases [91]. These potential mechanisms driving the beneficial effects of oligofructose may be due to upstream alterations of the gut microbiota [132].

Overall, oligofructose is one of the most promising dietary fibres extensively studied in randomized, double-blind, placebo-controlled crossover trials, which have contributed to substantial improvement of metabolic parameters in humans.

### 3.3. The Inter-Individual Responses to Prebiotics Require Personalised Nutrition Strategies

A substantial number of human dietary interventions have revealed that prebiotics can have highly variable effects in the metabolic health outcomes and the gut microbiota [142]. Indeed, the gut microbiota responds rapidly to changes in our diet, on a time scale of as little as 24 h, resulting in temporal fluctuations on time scales from hours to days [143,144]. Despite maintaining the dietary intervention, the gut microbiota can nearly return to its original baseline state for the remainder of the intervention [145,146]. This microbial resilience phenomenon is the result of long-term dietary habits that constitute the determinant force of an individual’s gut microbiota [147,148]. Hence, the gut microbiota exhibits considerable interpersonal variations in taxonomic composition and functions, regardless of individual health status [149,150]. Evidence reported that over 20% of the interpersonal variability of microbial signatures can be inferred from environmental factors associated with diet and lifestyle [145,151]. Geography is a major factor with dominant species in the gut showing a large degree of individual variation across cultures and continents [152]. The tremendous variability of microbial responses to dietary interventions has been correlated to a variety of confounding factors, including the use of medications (e.g., antibiotics, osmotic laxatives, female hormones, benzodiazepines, antidepressants, and antihistamines) and stool consistency, among others [153]. 

Differential clinical responses to prebiotics in adults can be more effective in some individuals, identified as responders, than in others, identified as non-responders (Figure 3). Responders are individuals with appropriate baseline commensal microbes for whom the prebiotic may confer a health benefit [21]. The reports of responders and non-responders in intervention studies encourage the determination of individual characteristics to better apprehend the efficacy of dietary intervention. The standardization of clinical protocols could provide details on the subjects enrolled in studies, including sex, age, ethnicity, diet, and the compositional and functional features of their gut microbiota. This gathering of in-depth clinical information would serve to predict individual responses to prebiotics [154,155]. For instance, a previous study characterised by high fibre and decreased energy intake illustrated the gene richness of the gut microbiome as a key microbial feature in pinpointing individuals who would respond efficiently to short-term dietary intervention [156]. This study highlighted that the positive response in obese patients was less effective in improving clinical phenotypes in individuals with lower microbial richness. Thus, the baseline of microbial diversity may be an important predictor of the response to diet-induced improvements in clinical responses to prebiotics [112]. Moreover, the magnitude of individual shifts induced by prebiotics can be restricted to selective responding bacterial species in the composition of the gut microbiota. The reasons for this individual variability in response to dietary intervention can be explained by the differences in enzyme capacity to utilise a prebiotic substrate [157] and/or can be due to the absence of keystone species [53]. For instance, previous studies have proposed profiling the gut microbiome for CAZymes as a key microbial feature that can profoundly influence how the host interacts with respect to diverse carbohydrate sources [158]. Referred to as CAZy-typing, the analysis of the repertoires of gut microbiome CAZyme-encoding genes can be used as an indicator to predict the metabolisation of prebiotics on an individual scale [159]. Indeed, the inconsistent responsiveness of individuals in dietary interventions can be explained by the absence of functional “guilds” able to access and utilise a carbon source [160]. 

Identifying specific characteristics of the gut microbiome will enable the development of individualised nutrition strategies [161] (Figure 3). Indeed, there is significant interest in targeted strategies to modulate microbial composition within hosts in a personalised approach to redirect microbial signatures towards health [155]. A recent personalised diet intervention successfully identified personal and microbiome characteristics to accurately predict personalised postprandial glycemic responses. This study revealed that exposure to specific dietary components modulated the composition of the gut microbiota that influenced host metabolic responses to lower postprandial glucose [162]. Therefore, the development of personalised diets that regulate blood sugar levels provides hope for further advancements in the control and treatment of disease. 

Future research on short-term and long-term dietary interventions could improve the characterisation responses of the gut microbiota to prebiotics to give us a deeper understanding of these interventions and their potential for precision application [163]. Health and microbiome measurability is required before clinical recommendations and can be made for dietary modulation of the gut microbiota to improve health and well-being. Although general recommendations may lay the foundation for nutritional guidelines, the path to a personalised nutrition approach will be an increasingly desirable research area in the near future. The development of precision nutritional strategies may need to be unique to each individual and tailor-made to each individual’s microbial composition.

## 4. Conclusions

Due to the variability of the responses of the gut microbiota, transforming theory into real-life host benefits is complicated. Future studies of the interactions between prebiotics and the gut microbiota is likely to rely on transdisciplinary fields, including evolution, ecology, microbiology, biomedicine, and computational biology. A better comprehension in the breakdown mechanisms of prebiotics increments requires fundamental research of the gut microbiota that supports the translation into clinical interventions, personalised nutrition approaches, and public recommendations. Collaborative efforts are still in their infancy and should be encouraged, taking into account public health issues.

## Figures and Tables

**Figure 1 nutrients-14-02096-f001:**
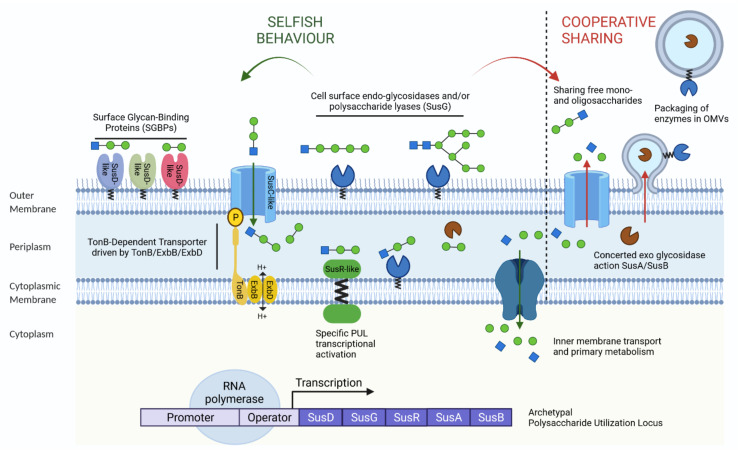
Nutrient acquisition strategies in two common trophic behaviours. The archetypal starch utilisation system (SUS) operon, a model system for starch uptake described in the commensal *Bacteroides thetaiotaomicron* at the origin of the polysaccharide utilisation locus (PUL). Intimately associated, the SusD cell-surface glycan binding proteins (SGBPs) initially adhere to and recruit the substrate from the outer membrane. The SusG endoglucanase proteins (GHs and PLs) hydrolyse starch into smaller malto-oligosaccharides that are further imported into the periplasm by the SusC Ton-B dependent transporter (TBDT). Carbohydrate-binding proteins and endoglucanase proteins vary substantially between PULs. Subsequently, oligosaccharides are catalysed into single sugars by the SusA and SusB exoglucosidases in the periplasmic space, before being imported into the cytoplasm for primary metabolism. In response to the presence of malto-oligosaccharides, the prototypic PUL regulator SusR protein senses degradation products to control the transcriptional activation of the PUL machinery. In the intestinal environment, the products of carbohydrate breakdown can be either slotted into primary metabolic pathways, which could be called selfish behaviour, or act for the public good. These latter show cooperative behaviour, where polymers and hydrolytic enzymes can simply diffuse or can be shared in outer membrane vesicles (OMVs). P, phosphorus; H^+^, hydrogen ions; SusA–SusD, SusG, SusR, SUS homologs; TonB, ExbB, ExbD, outer membrane receptors.

**Figure 2 nutrients-14-02096-f002:**
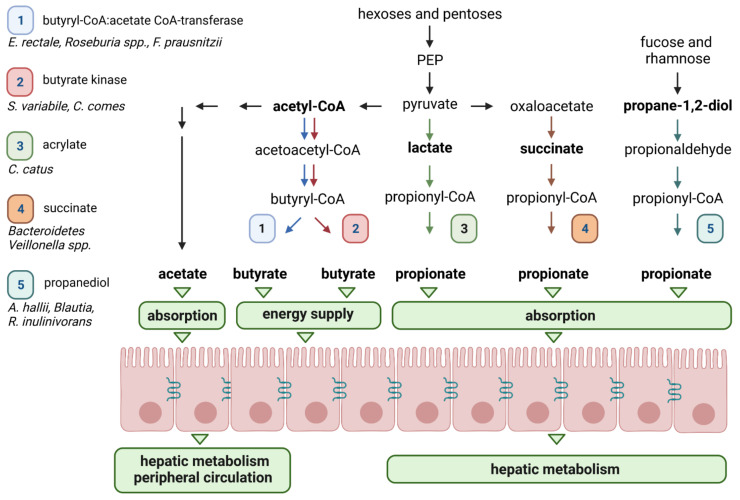
Short-chain fatty acid (SCFA) biosynthesis pathways by the gut microbiota. The different pathways involved in SCFA production are presented for acetate, butyrate, and propionate. SCFA-producing bacteria for each pathway are also shown. Acetate is mainly produced in the gut from pyruvate via acetyl-coenzyme A (CoA). Three pathways have been described for propionate synthesis, namely acrylate, succinate, and propanediol. The first two start from phosphoenolpyruvate (PEP) and the latter uses deoxysugars, such as fucose and rhamnose. Butyrate is produced via two pathways: the butyryl-CoA: acetate CoA-transferase and the butyrate kinase.

**Figure 3 nutrients-14-02096-f003:**
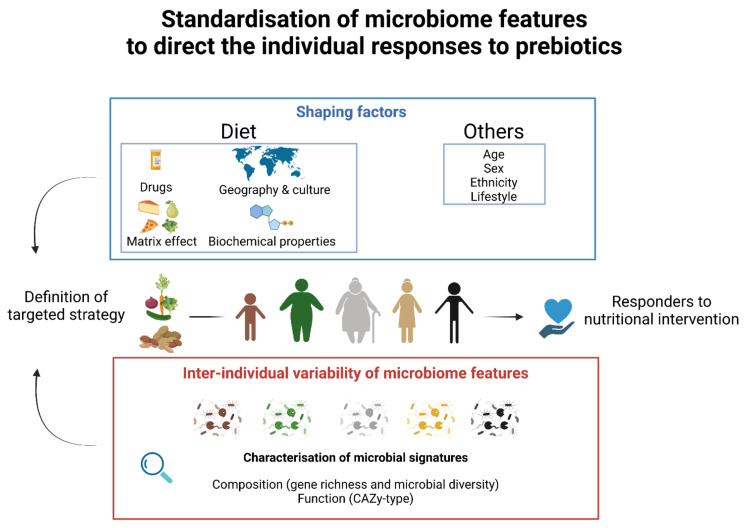
The determination of individual characteristics to direct an efficient prebiotic intervention. Individual and environmental shaping factors and interindividual variability of microbiomes modulate differential clinical responses to prebiotics. The description of microbiome signatures at the compositional and functional levels can provide insights to define a targeted nutritional strategy. A standardisation of multi-criteria aims to match the prebiotic intervention with individuals that would likely respond efficiently.

**Table 1 nutrients-14-02096-t001:** The prebiotic concept over the years. Consisting of a panel of academic and industrial experts, the ISAPP regularly convenes to state the definition and scope of prebiotics. New considerations are gradually included along with scientific research progress, consumer interest, and technological innovations from industrial scientists.

Evolution of the Prebiotic Concept	Additional Considerations from the Previous Definition	Ingredients Incrementally Considered as Prebiotics
“non-digestible food ingredient that beneficially affects the host by selectively stimulating the growth and/or activity of one or a limited number of bacteria in the colon, and thus improves host health” [22]	None	FOS
“selectively fermented ingredient that allows specific changes, both in the composition and/or activity in the gastrointestinal microbiota that confer benefits upon host well-being and health” [20]	(a) non-digestibility (b) fermentation by intestinal microflora (c) selective stimulation of the growth and/or activity of intestinal bacteria	Inulin tGOS Lactulose Candidates * are listed: IMO, lactosucrose, SOS, XOS, GlOS, and other compounds
NB: there is no new definition of a prebiotic, but rather a validation and an expansion of the prebiotic concept [26]	(a) nature of the prebiotics (b) dose-effect relation (c) animals and humans	None
NB: there is no new definition of a prebiotic, but rather a validation and an expansion of the prebiotic concept [23]	(a) increase in the genus *Bifidobacterium* as a marker of intestinal health (b) selectivity of other genera or species than bifidobacteria (e.g., butyrate-producing bacteria) (c) beneficial effects in the colon and the whole body	None
“a non-digestible compound that, through its metabolisation by microorganisms in the gut, modulates composition and/or activity of the gut microbiota, thus conferring a beneficial physiological effect on the host” [24]	(a) anatomical restriction to the gut (b) requirement or not of fermentation (c) restriction only to carbohydrates (d) requirement or not of microbiota modulation (possibility of having other direct positive effects)	HMO Candidates * are listed: RS, pectin, AX, whole grains, various dietary fibres and other non-carbohydrates such as polyphenols
“a substrate that is selectively utilised by host micro-organisms conferring a health benefit” [19]	(a) microbes targeted by prebiotics should be health-promoting bacteria without specifying which ones (b) effect is no longer limited to the microbial community of the gastrointestinal system associated with humans or animals (c) importance in describing selective bacterial metabolism and assessing microbial function and composition in reproducible randomized controlled studies that establish the direct link between prebiotics and health in the specific target host	Candidates * are listed: HMO, MOS, and other non-carbohydrates such as polyphenols, CLA, and PUFA

Abbreviations: ISAPP, international scientific association of probiotics and prebiotics; NB, nota bene; FOS, fructo-oligosaccharides; tGOS, trans-galacto-oligosaccharides; IMO, isomalto-oligosaccharides; SOS, soya-oligosaccharides; XOS, xylo- oligosaccharides; GlOS, gluco-oligosaccharides; HMO, human milk oligosaccharides; RS, resistant starches; AX, arabinoxylans; MOS, mannan-oligosaccharides; PUFA, polyunsaturated fatty acids; CLA, conjugates linoleic acids * The prebiotic potential of candidate compounds has been investigated. However, scientific evidence is too sparse at the time to demonstrate any prebiotic effects.

## Data Availability

Not applicable.

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
