# Peer review of "Prebiotics and the Human Gut Microbiota: From Breakdown Mechanisms to the Impact on Metabolic Health"

_nutrients, 2022, doi:10.3390/nu14102096_

Round 1

Reviewer 1 Report

I am grateful for the opportunity to review this manuscript, and I enjoyed reading it. The authors did an extensive literature review so references of this manuscript are more than appropriate. I believe this is a promising study field and this article will be of interest to a broad readership. Figures are also excellent.

I only have a few minor suggestions for the authors:

 1. add study design to the title

2. exclude reference from the abstract

3. add a conclusion on the findings to abstract

4. why are Gibson et al and other Gibson reference cited using a different style

5. ISAPP is twice described in the text, please add it under the table 1, not in the name

6. page 13 Clinical evidence of prebiotic effects please make this first part as at least 3 paragraphs so it is easier to read

7. I would appreciate few sentences for limitations of your study

Author Response

We are grateful to the referee for his/her comments on our review. Please find below our modifications according to referee’s requests.

  1. According to the comment regarding the title of the study design. We have add to title that this review focuses on the human gut microbiota (title page).
  2. Moreover, we have excluded the reference from the abstract (title page).
  3. In addition, we have completed the abstract with one sentence conclusion (title page).
  4. We have edited the reference citation style in the content.
  5. We have corrected the redundancy of the ISAPP description (pages 2 & 3).
  6. The paragraph "clinical evidence" was divided into several parts (page 13).
  7. Finally, we added few sentences for the limitations of the study (page 16).

Reviewer 2 Report

This manuscript is the research about " Prebiotics and the gut microbiota: from breakdown mechanisms to the health impact in metabolic-related diseases" submitted by Cassandre Bedu-Ferrari , Paul Biscarrat , Philippe Langella , Claire Cherbuy *. The manuscript writes very well. But some points need to be improved.

  1. The title must be corrected to match the content of the article
  2. Please increase the possible location of nutrient acquisition strategies for the distribution of microbial communities in the gut
  3. Prebiotics "IMO, SOS, Glos, MOS" in Table 1, there is no literature description in the content, why?
  4. Please add the role of SCFA in metabolic diseases or why they affect microbial colonies
  5. The conclusion should be improved
  6. Inconsistent citation formats in the content should be improved
  7. Some citations do not appear in the content, please check again

Author Response

We are grateful to the referee for his/her comments on our review. Please find below our modifications according to referee’s requests.

  1. The title was corrected to match the content of the article (title page).

  2. The principal location for prebiotic metabolisation and nutrient acquisition strategies is the distal part of the gut. We have added few words to clarify this point (page 8).

  3. References regarding the prebiotics IMO, SOS, GlOS, MOS are specified in the Table I (page 3). Given the abundance of literature, it is not possible to develop the content for all potential prebiotic candidates. We have preferred in the present article to focus only on complex carbohydrates that represent the best-studied class of prebiotic till to now, which mainly include ITF and GOS.

  4. We had the role of SCFA in metabolic diseases (page 13).

  5. Following the request of the other reviewer, the conclusion has been slightly changed (page 16).

  6. We have edited the reference citation style in the whole content.

  7. We have corrected the citations in the whole content.